# Protozoan and Rickettsial Pathogens in Ticks Collected from Infested Cattle from Turkey

**DOI:** 10.3390/pathogens11050500

**Published:** 2022-04-22

**Authors:** Shengwei Ji, Onur Ceylan, Zhuowei Ma, Eloiza May Galon, Iqra Zafar, Hang Li, Yae Hasegawa, Mutlu Sevinc, Tatsunori Masatani, Aiko Iguchi, Osamu Kawase, Rika Umemiya-Shirafuji, Masahito Asada, Ferda Sevinc, Xuenan Xuan

**Affiliations:** 1National Research Center for Protozoan Diseases, Obihiro University of Agriculture and Veterinary Medicine, Obihiro 080-8555, Japan; jishengwei0903@hotmail.com (S.J.); mazhuowei1994@gmail.com (Z.M.); eloizagalon@gmail.com (E.M.G.); eekrawahla@hotmail.com (I.Z.); lihang-2020@hotmail.com (H.L.); y13665trp@gmail.com (Y.H.); umemiya@obihiro.ac.jp (R.U.-S.); masada@obihiro.ac.jp (M.A.); 2Department of Parasitology, Faculty of Veterinary Medicine, Selcuk University, Konya 42250, Turkey; onurceylan@selcuk.edu.tr; 3Department of Internal Medicine, Faculty of Veterinary Medicine, Selcuk University, Konya 42250, Turkey; msevinc@selcuk.edu.tr; 4Faculty of Applied Biological Sciences, Gifu University, Gifu 501-1193, Japan; mstn@gifu-u.ac.jp; 5Joint Department of Veterinary Medicine, Faculty of Agriculture, Tottori University, Tottori 680-8550, Japan; iguchi@tottori-u.ac.jp; 6Department of Biology, Premedical Sciences, Dokkyo Medical University, Tochigi 321-0293, Japan; osamuk@dokkyomed.ac.jp

**Keywords:** tick-borne pathogens, tick species, rickettsial pathogens, Turkey

## Abstract

Diseases caused by tick-transmitted pathogens including bacteria, viruses, and protozoa are of veterinary and medical importance, especially in tropical and subtropical regions including Turkey. Hence, molecular surveillance of tick-borne diseases will improve the understanding of their distribution towards effective control. This study aimed to investigate the presence and perform molecular characterization of *Babesia* sp., *Theileria* sp., *Anaplasma* sp., *Ehrlichia* sp., and *Rickettsia* sp. in tick species collected from cattle in five provinces of Turkey. A total of 277 adult ticks (males and females) were collected. After microscopic identification, tick pools were generated according to tick species, host animal, and sampling sites prior to DNA extraction. Molecular identification of the tick species was conducted through PCR assays. Out of 90 DNA pools, 57.8% (52/90) were detected to harbor at least 1 pathogen. The most frequently-detected pathogens were *Babesia bovis*, with a minimum detection rate of 7.9%, followed by *Ehrlichia* sp. (7.2%), *Theileria annulata* (5.8%), *Coxiella* sp. (3.3%), *Anaplasma* *marginale* (2.5%), *Rickettsia* sp. (2.5%), and *B. occultans* (0.7%). *Rickettsia* sp. identified in this study include *Candidatus* Rickettsia barbariae, *R. aeschlimannii*, and *Rickettsia* sp. Chad. All sequences obtained from this study showed 99.05–100% nucleotide identity with those deposited in GenBank (query cover range: 89–100%). This is the first molecular detection of *Rickettsia* sp. Chad, a variant of Astrakhan fever rickettsia, in Turkey. Results from this survey provide a reference for the distribution of ticks and tick-borne pathogens in cattle and expand the knowledge of tick-borne diseases in Turkey.

## 1. Introduction

Ticks and tick-borne diseases (TBDs) significantly affect livestock production in many countries of the world. They are considered second only to mosquitoes in importance as vectors of disease agents causing great impact on animal and human health [1]. The global distribution of ticks and TBDs have led to considerable economic losses, such as loss of productivity, decreased meat and milk production, and death of livestock [2,3,4,5]. To date, 731 species of Ixodidae (hard ticks), 216 species of Argasidae (soft ticks), and 1 species of Nuttalliellidae (*Nuttalliella namaqua*) have been described worldwide [6]. Due to suitable climatic conditions and the large amounts of wild and domestic animals, a total number of 51 tick species have been identified in Turkey (43 from the family Ixodidae and 8 from the family Argasidae). *Dermacentor*, *Haemaphysalis*, *Hyalomma*, *Ixodes*, and *Rhipicephalus* ticks are widely present throughout Turkey [7,8,9,10].

In Turkey, 27% of farm animals are cattle, with approximately 90% of milk and meat produced from bovines [11]. However, TBDs have a significant impact on the economy and animal health. The epidemiology of TBDs in Turkey had been reported previously [12,13,14], but there is still limited data about pathogens carried or transmitted by ticks. Therefore, the aims of this study were to investigate the presence of and genetically characterize the pathogens harbored by ticks collected from cattle to improve understanding of their distribution towards effective control in Turkey.

## 2. Results

### 2.1. Tick Species Identification

In total, 277 ticks were collected from 22 severely infested cattle in provinces of Diyarbakır (n = 2), Gaziantep (n = 4), Kahramanmaraş (n = 4), Karaman (n = 9), and Şanlıurfa (n = 3) (Figure 1). Morphological observation-based identification revealed that the 277 ticks belong to two tick genera, namely *Hyalomma* and *Rhipicephalus*. The five species were identified as *Hyalomma marginatum* (n = 2), *H. excavatum* (n = 203), *H. anatolicum* (n = 5), *Rhipicephalus turanicus* (n = 60), and *Rh. bursa* (n = 7) (Figure 1). The length of the obtained tick mitochondrial *16S rDNA* sequences varied from 448 to 453 bp and the BLASTn analysis of the *H. anatolicum*, *H. excavatum*, *H. marginatum*, *Rh. turanicus*, and *Rh. bursa* sequences in this study showed percent identity ranging from 99.33 to 100% (query cover range: 89–100%) with deposited sequences from GenBank (Appendix A).

### 2.2. Microorganisms Detected in Ticks and Minimum Detection Rates

A total of 57.8% of tick pools (52/90) harbored at least one pathogen (Table 1). Seven pathogens, namely *Anaplasma marginale* (7), *Babesia bovis* (22), *B. occultans* (2), *Coxiella* sp. (9), *Ehrlichia* sp. (20), *Rickettsia* sp. (7), and *Theileria annulata* (16) were detected in the tick pools (Table 1). The most frequently-detected pathogens in this study (Table 1) were *B. bovis* (minimum detection rate 7.9%, 22 positive pools), followed by *Ehrlichia* sp. (minimum detection rate 7.2%,20 positive pools), *T. annulata* (minimum detection rate 5.8%, 16 positive pools), *Coxiella* sp. (minimum detection rate 3.3%, 9 positive pools), *A. marginale* and *Rickettsia* spp. (minimum detection rate 2.5%, 7 positive pools each), and *B. occultans* (minimum detection rate 0.7%, 2 positive pools).

For *Rickettsia* sp., 7 pools were positive when screened using the 16S rRNA assay, whereas 5 of 7 were positive when screened for *gltA* and *ompA* genes. Three samples positive for all genes of *Rickettsia* and two samples positive for 16S rRNA only were selected for sequencing. In the case of *Anaplasma/Ehrlichia*, all samples that showed an amplicon using the 16S rRNA assay were positive using *A. marginale msp4* and/or *E. ruminantium pCS20* assays, thus, amplicons from the latter assays only were sequenced.

### 2.3. Analyses of Tick-Borne Pathogen DNA Sequences

Positive samples of each pathogen were randomly selected as representative for sequencing and analysis. The NCBI BLASTn analysis of *A. marginale* (OL408894; *H*. *excavatum*), *B. bovis* (OL408893; *H*. *excavatum*), and *B. occultans* (OL377855; *H*. *excavatum*) sequences obtained from this study showed 100% identity with the reference sequences MF377455 (Turkey cattle), MN870661 (Egypt cattle), and KP745626 (Turkey cattle), respectively. The *Coxiella* sp. sequence (OL413002; *Rh. turanicus*) identified in this study showed 99.66% identity with *Candidatus* Coxiella mudrowiae (CP011126), which was detected in a *Rh. turanicus* tick in Israel. BLASTn analysis hits of *Rickettsia* sp. (16S rRNA, *gltA*, and *ompA*) included three main species including *R. aeschlimannii* (OL377895, OM541406, OM717963 and OM717960), *Candidatus* Rickettsia barbariae (OL377896, OM541407, OM717964, and OM717961), and *Rickettsia* sp. Chad (OM541405, OM717962, and OM717959). OL377895, and OM541406 were detected in different *H. excavatum* pools which were collected from different cattle in Kahramanmaraş. Meanwhile, OL377896 was detected in *H. excavatum* in Kahramanmaraş, while OM541407 was detected in *Rh. turanicus* in Karaman. The BLASTn analysis showed 99.29–100% (query cover range: 95–100%) with reference strains (Table 2). Furthermore, *T. annulata* (OL408892) shared the highest percent identity (99.07%) with a dog isolate from Tunisia (KX130956) (Table 2).

### 2.4. Phylogenetic Analyses of Rickettsial Agents

The *A. marginale msp4* (OL408894) sequence from this study was found in the same clade with cattle isolates from Turkey (MF377455), Italy (KF739428), Algeria (KX179906), and Tunisia (KJ512170) (Figure 2). Meanwhile, the *Ehrlichia* sp. *pCS20* sequence obtained in this study clustered with KT362172, an isolate from *Rh. (Boophilus) microplus* in China (Figure 3). Phylogenetic trees inferred from 16S rRNA, *gltA*, and *ompA* genes of *Rickettsia* sp. are shown in Figure 4. The *Rickettsia* 16S rRNA sequences were grouped in two clades (Figure 4A). Two sequences (OM541406 and OL377895) created a polyphyletic group with other *R. aeschlimannii* and *R. massiliae* sequences, while the remaining sequences (OM541405, OM541407, and OL377896) were grouped into *R. conorii* complex clade. Interestingly, OM541405 formed a well-supported branch with *Rickettsia* sp. isolated from a human with clinical Astrakhan fever in Chad. Phylogenetic analysis of *gltA* (Figure 4B) and *ompA* (Figure 4C) sequences further confirmed that OM717962 and OM717959 were most closely related to *Rickettsia* sp. Chad. Meanwhile, OM717963 and OM717960 were in the same group of *R. aeschlimannii*, whereas OM717964 and OM717961 were in the same group of *Candidatus* Rickettsia barbariae, respectively (Figure 4).

## 3. Discussion

Tick-borne protozoan and rickettsial diseases are major health problems for cattle and cause economic losses worldwide [4,14,15]. In recent years, studies related to ticks and TBDs have increased in Turkey. Understanding the complex relationships between ticks and pathogens is important to public health in mitigating tick-borne pathogen (TBP) transmission. In this study, we performed a molecular survey of tick-transmitted pathogens in ticks obtained from cattle in Diyarbakır, Gaziantep, Kahramanmaraş, Karaman, and Şanlıurfa provinces located in the south of Turkey. Overall, seven tick-associated microorganisms, including *A. marginale*, *B. bovis*, *B. occultans*, *Coxiella* sp., *Ehrlichia* sp., *Rickettsia* sp. (*Candidatus* R. barbariae, *R. aeschlimannii*, and *Rickettsia* sp. Chad), and *T. annulata* were found in four tick species identified, namely *H. excavatum*, *H. anatolicum*, *Rh. turanicus*, and *Rh. bursa*.

Six *Anaplasma* sp., including *A. bovis*, *A. centrale*, *A. marginale*, *A. ovis*, *A. platys*, and *A. phagocytophilum*, cause a range of diseases in humans and other vertebrates [16]. The most common agent for bovine anaplasmosis is *A. marginale,* which is highly pathogenic and can be fatal in susceptible cattle [17]. Major surface protein 4 (MSP4) is an important protein, which is considered as a stable marker for genetic characterization of *A. marginale* strains [18,19]. In this study, a molecular survey based on the PCR amplification of the target gene *msp4* was performed and *A. marginale* was detected in *H. excavatum* and *Rh. turanicus* ticks with a minimum detection rate of 2.5%. Previously, *A. marginale* has been reported in many regions of Turkey, which has been confirmed as the most prevalent tick-borne pathogen in cattle [10,15,20,21,22,23,24]. However, *A. marginale* infection in ticks was rarely reported in Turkey. The phylogenetic analysis of *A. marginale* revealed that the *msp4* sequence obtained from this study was highly similar to isolates from other Mediterranean countries. This result demonstrates the utility of *A. marginale msp4* as a marker for phylogeographic characterization of *A. marginale* isolates from the field [18].

Bovine babesiosis is one of the most important TBDs of cattle in Turkey and has been reported in all Turkish provinces [11]. The most important causative agents of bovine babesiosis are *B. bovis* and *B. bigemina*, which are transmitted by *Rhipicephalus* ticks. The minimum detection rate of *B. bovis* in ticks was 7.9% and was only detected in ticks from Gaziantep and Kahramanmaraş. Interestingly, we did not identify ticks harboring *B. bigemina*. In Turkey, *B. bigemina* is the most frequent *Babesia* species and is transmitted by *Rh. annulatus* ticks [10]. Such results may be explained by the limited number of collected tick species and the number of samples in this study. Meanwhile, *B. occultans* was first described in South Africa and Nigeria and has been identified in *Hyalomma* ticks [25,26]. Of the 90 pools, 2 (minimum detection rate 0.7%) pools from Şanlıurfa and Karaman were found positive for *B. occultans*, which was carried by *H. excavatum* in this study. Previous reports have proved the presence of *B. occultans* in Turkey [27] and found that it was not only carried by *Hyalomma* ticks, but was also observed in *Rh. turanicus* ticks [22].

*Coxiella burnetii* causes Q fever, a zoonotic disease which threatens human health and commonly causes abortions in cattle and sheep. In the present study, the *Coxiella* sp. positive tick species were *H. excavatum*, *Rh. turanicus*, and *Rh. bursa*. The presence of *Coxiella* sp. in ticks has been reported previously in Turkey wherein it was detected in *H. marginatum*, *H. anatolicum excavatum*, *H. detritum*, and *Rh. annulatus* ticks [28]. In another study, *Coxiella* sp. DNA was detected in *H. excavatum*, *Rh. turanicus*, and *Rh. bursa* ticks [29]. This suggests that *Coxiella* sp. is widespread in a variety of *Hyalomma* and *Rhipicephalus* species in Turkey. Additional studies are warranted to determine the presence of *C. burnetii* in the tick samples from this study.

*Ehrlichia ruminantium* causes heartwater disease in cattle and the main vector is the *Amblyomma* tick, particularly *A. hebraeum* and *A. variegatum* [30]. The partial sequences of *E. ruminantium* were reported previously in *Rh. evertsi evertsi*, *H. truncatum*, and *H. marginatum* ticks [31]. In this study, we detected *Ehrlichia* sp. in different tick species, including *H. anatolicum*, *H. excavatum*, *Rh. turanicus*, and *Rh. bursa*, suggesting that *Ehrlichia* sp. may be carried by *Hyalomma* and *Rhipicephalus* ticks. So far, only a few studies have documented the detection of *Ehrlichia* in ticks. Although *Ehrlichia* sp. was detected in *H. excavatum* and *Rh. bursa* in studies conducted in Turkey, that *Ehrlichia* sp. sequence found was similar to *E. canis* and *Ehrlichia* sp. Omatjenne strain [21,22]. The present study used an *E. ruminantium*-specific PCR assay, but the obtained *Ehrlichia* sp. *pCS20* gene sequence has a 99.28% identity with the *Ehrlichia* sp. (KT362172) sequence, previously reported in Hunan province, China [32]. Further studies are needed to confirm the species of the detected *Ehrlichia*, to investigate its presence in the blood samples of bovines where the positive ticks were collected from, and assess its significance for animal health.

Three species of *Rickettsia* were confirmed by phylogenetic analysis based on the 16S rRNA, *gltA*, and *ompA* genes in this study, including *Candidatus* Rickettsia barbariae (OM541407, OM717964, and OM717961 from one sample), *R. aeschlimannii* (OM541406, OM717963 and OM717960 from one sample), and *Rickettsia* sp. Chad (OM541405, OM717962, and OM717959 from one sample). Our findings confirmed the presence of *Candidatus* Rickettsia barbariae in Turkey, which is in concordance with a previous report [33]. This rickettsial agent’s DNA was firstly detected in *Rh. turanicus* ticks collected from European hare (*Lepus europaeus*) in 2020 [34]. The present study revealed the presence of *Candidatus* Rickettsia barbariae in ticks and reported the presence of this agent in Turkey for the second time. Moreover, the *Rickettsia* sp. Chad isolated from a patient was reported previously in a clinical Astrakhan fever case [35]. There is no information about the vector of *Rickettsia* sp. Chad. We have detected *Rickettsia* sp. Chad in *Rh. turanicus* in this study. Although there are only a few reports of *Rickettsia* sp. Chad infections in humans, our result suggests that *Rickettsia* sp. Chad infection may be a potential threat for humans, which manifests as fever and maculopapulous rash after infection [35]. As it was detected in ticks of cattle, further studies regarding its infectivity on cattle should be conducted.

Theileriosis is another tick-borne protozoan disease of cattle. The causative agents are transmitted from an infected animal to others by transstadial transmission via ticks. *Theileria annulata* transmitted by *Hyalomma* is the main TBP affecting cattle in Turkey [10]. The current study detected *T. annulata* infection in *H. excavatum*, *H. anatolicum*, and *Rh. turanicus* with a minimum detection rate of 5.8%. The main vectors of *T. annulata* are some species in the genus *Hyalomma*, and there are studies conducted in Turkey reporting that *T. annulata* was detected in *H. excavatum* [22,33]. *Rhipicephalus turanicus* harboring *T. annulata* was also reported in a previous study [36]. Therefore, it is necessary to evaluate whether *Rh. turanicus* will be a risk for *T. annulata* transmission in Turkey.

The detection of *Coxiella* sp. from ticks in this study despite using specific PCR primers, coupled with the results in BLASTn analysis, indicates that this species is an endosymbiont. Endosymbionts are non-pathogenic microorganisms that have evolved in the carriers or hosts either as obligate or facultative symbionts [37]. Ticks were reported previously to be highly infected with endosymbionts [38,39]. In this study, an isolate highly similar to *Candidatus* Coxiella mudrowiae, a known *Coxiella* endosymbiont, was detected in *Rh. turanicus* ticks, as it was similarly described in a previous study [40]. The extent to which the endosymbionts of ticks in Turkey affect transmission of other TBPs needs to be studied in the future.

In the study, a large number of single or multiple pathogenic microorganisms’ DNA was detected in cattle tick pools. It is difficult to determine whether these microorganisms originated from ticks or from hosts blood since all ticks were collected while sucking blood from cattle. Karim et al. [38] reported that the characterization of multiple infections in ticks poses a major scientific challenge for understanding the epidemiology of tick-borne infectious diseases. However, since the tick DNA samples in this study were not extracted from each individual ticks but from 1–10 ticks collected in pools, this limits the discussion of single and coinfection status. Additionally, our study does not discuss the co-transmission of many microorganisms from ticks to cattle, and further studies are needed to obtain information about potential tick-borne pathogens and the dynamics of tick-borne infections in a region [38,39].

The limitation of this study was that it was carried out on a restricted number of cattle. The collection of ticks at a certain time of the year and the lack of periodic tick collection according to the seasons are considered as other limiting factors of this study. The study is also limited in choosing sampling locations and, thus, could not cover the whole of Turkey to provide a better representation of tick and tick-borne pathogens. Another limiting factor is the inability to obtain blood from infested cattle. Although we were aware of the importance of this, it was outside the scope of the present study.

In conclusion, the results obtained from the present study will contribute to the present knowledge on the distribution of ticks and the possible TBPs they may carry. Further investigation for the vector competence of tick species in transmitting these pathogens and more research with a greater number of ticks, blood samples, and sampling locations should be conducted in order to fully assess the risk factors for tick-transmitted livestock diseases in Turkey.

## 4. Materials and Methods

### 4.1. Ethical Statements

The owners of the cattle were informed about the study, and their approvals were obtained for tick collection. Tick sampling, as well as sample processing, were carried out according to the ethical guidelines permitted by Obihiro University of Agriculture and Veterinary Medicine (Permit for DNA experiment: 1723-4 and 1724-4; Pathogen: 201712-5).

### 4.2. Tick Samples and DNA Extraction

Tick samples were collected from March to June 2013. Seventy apparently healthy cattle were examined. Tick infestations (1–45 ticks per animal) were determined in 22 (31.43%) of these cattle. The study consisted of 277 adult ticks, which were collected in 5 provinces (Diyarbakır, Gaziantep, Karaman, Kahramanmaraş, Şanlıurfa) of Turkey. All ticks were removed from cattle skin with the consent of the livestock owner, and care was taken to minimize animal discomfort. The tick samples were kept in 70% ethanol prior to identification using a binocular microscope (Olympus SZX16, Tokyo, Japan) based on standard taxonomic keys [41,42]. Tick identification was done at the National Research Center for Protozoan Diseases, Obihiro University of Agriculture and Veterinary Medicine, Japan. After microscopic identification, the tick samples were pooled according to species, gender, and sampling site. Ninety pools were generated based on sample size (1 to 10 ticks per pool). Tick DNA extraction was done as previously described [43] by using a QIAamp DNA Mini Kit (QIAGEN, Hilden, Germany) according to the manufacturer’s instructions. The extracted DNA was eluted with 50 μL of double-distilled water and stored at −30 °C until use.

### 4.3. Molecular Characterization of Ticks

To further confirm the tick species, molecular identification of the tick species was conducted through PCR assays. A primer set targeting the *16S rDNA* of ticks was used, followed by cloning and sequencing of amplicons (Appendix A). DNA amplification was done with a 10 μL PCR reaction volume containing 1 μL of 10 × Thermopol^®^ Buffer (New England Biolabs, Ipswich, MA, USA), 0.2 μL of 10 mM dNTP mix (New England Biolabs, Ipswich, MA, USA), 0.2 μL of 10 μM forward and reverse primers, 0.05 μL of Taq polymerase (New England Biolabs, Ipswich, MA, USA), 1 μL of DNA sample, and 7.35 μL of double distilled water. The thermocycling conditions consisted of an initial denaturation (95 °C for 2 min); 30 cycles of denaturation (95 °C for 30 s); annealing (52 °C for 30 s) and extension (68 °C for 1 min), then final extension (68 °C for 5 min) [44]. PCR products were checked by electrophoresis on 1.5% agarose gels, stained with ethidium bromide, and visualized under UV light. All amplicons were extracted from gels using a Gel Extraction Kit (QIAGEN, Hilden, Germany) and sequenced.

### 4.4. Detection and Characterization of Tick-Borne Pathogens

A total of 90 tick DNA pools were generated in this study. All tick pool samples were primarily screened using primers targeting the genes 18S rRNA (V4 hypervariable region) for *Babesia* and *Theileria* [45], 16S rRNA for *Anaplasma* and *Ehrlichia* [46], and 16S rRNA for *Rickettsia* [47], as listed in Appendix A. The positive samples based on these TBP assays were selected for species-specific detection. Partial sequences of *B. bovis spherical body protein 4* (*sbp4*) [48], *B. bigemina rhoptry-associated protein 1a* (*RAP1a*) [48], *T. orientalis major piroplasm surface protein* (*MPSP*) [49], *T. annulata merozoite surface antigen 1* (*Tams-1*) [50], *Anaplasma marginale major surface protein 4* (*msp4*) [51], *Ehrlichia pCS20* [52], *C. burnetii* 16S rRNA [53], and *Rickettsia* citrate synthase *gltA* and *ompA* [54,55] genes were amplified. The primers used are shown in Appendix A. The DNA amplification and PCR cycling for all pathogens followed the same conditions as aforementioned, except for the annealing temperatures, wherein those documented in referenced publications were used. PCR products were checked and purified as described above. The samples were further confirmed by sequencing.

### 4.5. Cloning and Sequencing

The amplicons (tick *16S rDNA* or pathogen DNA) were cloned in pGEM-T Easy Vector and sequenced as described [56] using the Big Dye Terminator v3.1 Cycle Sequencing Kit (Applied Biosystems, Waltham, MA, USA) and the ABI PRISM 3100 genetic analyzer (Applied Biosystems). The GenBank accession numbers are shown in Table 2 and Appendix A.

### 4.6. Sequence Alignment and Phylogenetic Analyses

The phylogenetic analyses of the sequences obtained from the PCRs specific for *A. marginale*, *Ehrlichia* sp., and *Rickettsia* sp. were performed subsequently, as rickettsial pathogens are relatively rarely studied in Turkey. Sequenced DNA samples were analyzed using the BLASTn tool of NCBI GenBank database and Clustal X program. In addition, phylogenetic analyses were inferred using MEGA version 7.0 software. The maximum likelihood method was employed as the method for tree construction because of its use of complex models to simulate biological reality and infer sequence evolution [57]. The sequences included in the phylogenetic analysis were chosen based on the BLASTn search results and geographical origin. Bootstrap analysis was performed with 1000 replicates to estimate the confidence of branching patterns of the tree.

## Figures and Tables

**Figure 1 pathogens-11-00500-f001:**
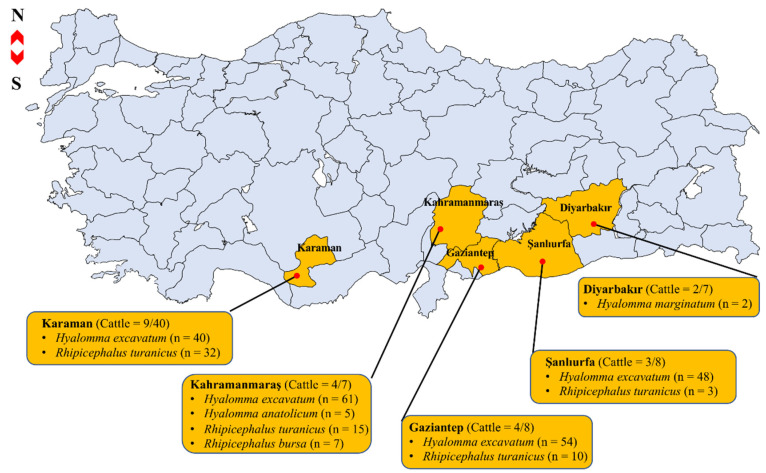
Map of sample collection sites in Turkey showing numbers of cattle examined (no. infested/total) and numbers of ticks of each species collected.

**Figure 2 pathogens-11-00500-f002:**
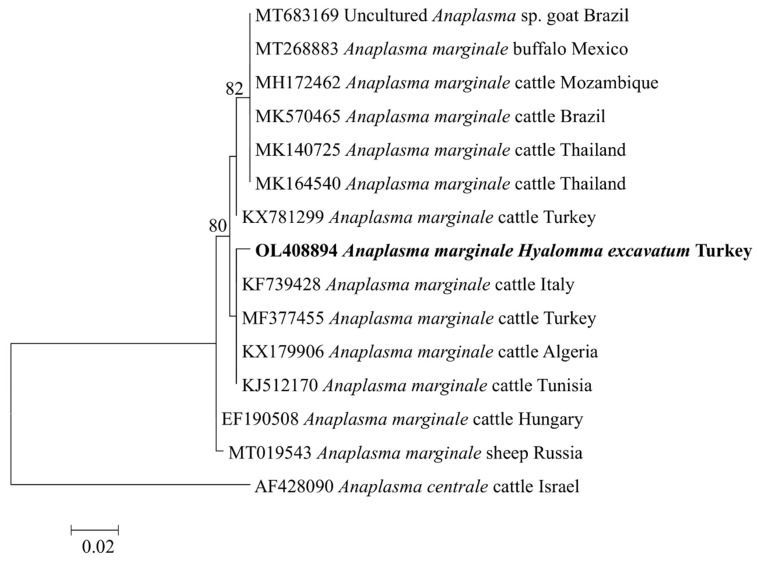
Phylogenetic analysis of *Anaplasma marginale* based on *msp4*. The sequences determined in this study are shown in bold font. Numbers at the nodes represent the percentage of occurrence of clades in 1000 bootstrap replicates of the taxa.

**Figure 3 pathogens-11-00500-f003:**
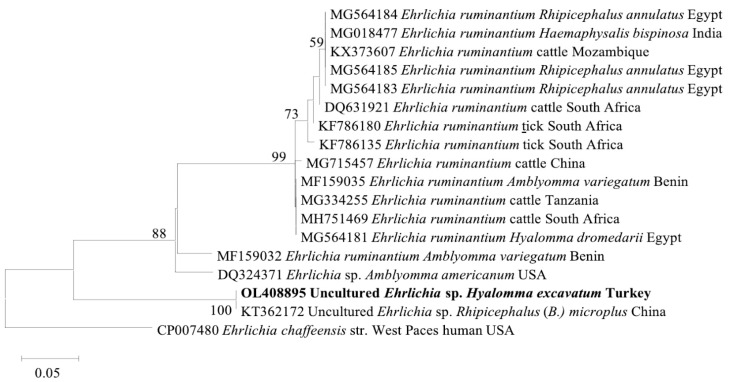
Phylogenetic analysis of *Ehrlichia* sp. based on *pCS20*. The sequences in bold font are from this study. The numbers at the nodes represent the percentage of occurrence of clades in 1000 bootstrap replicates of the taxa.

**Figure 4 pathogens-11-00500-f004:**
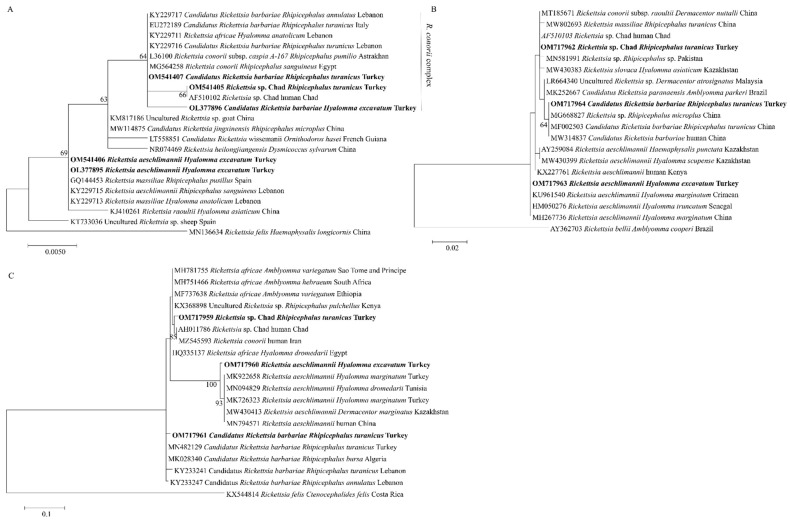
Phylogenetic analysis of *Rickettsia* sp. based on 16S rRNA (**A**), *gltA* (**B**) and *ompA* (**C**) detected in ticks collected from Turkey. The sequences determined in this study are shown in bold font. The numbers at the nodes represent the percentage of occurrence of clades in 1000 bootstrap replicates of the taxa.

**Table 1 pathogens-11-00500-t001:** Detection of microorganisms in tick pools based on provinces and total minimum detection rates.

Province	Tick Species (Positive Pools/Total Pools)	Microorganism Detected (No. of Pools)	Minimum Detection Rate
Gaziantep	*Hyalomma excavatum* (11/17)	*Anaplasma marginale* (4)	
		*Babesia bovis* (8)	
		*Ehrlichia* sp. (3)	
		*Theileria annulata* (4)	*A. marginale* 2.5% (7/277)
	*Rhipicephalus turanicus* (1/8)	*Ehrlichia* sp. (1)	
Kahramanmaraş	*H*. *excavatum* (16/16)	*A*. *marginale* (1)	
		*B*. *bovis* (11)	
		*Coxiella* sp. (6)	*B. bovis* 7.9% (22/277)
		*Ehrlichia* sp. (4)	
		*Rickettsia* sp. (3)	
		*T*. *annulata* (4)	
	*Rh*. *turanicus* (4/6)	*A*. *marginale* (1)	*B*. *occultans* 0.7% (2/277)
		*B. bovis* (1)	
		*Coxiella* sp. (2)	
		*Ehrlichia* sp. (1)	
		*Rickettsia* sp. (2)	*Coxiella* sp. 3.3% (9/277)
	*Rh*. *bursa* (2/2)	*B*. *bovis* (1)	
		*Coxiella* sp. (1)	
		*Ehrlichia* sp. (1)	
	*H*. *anatolicum* (2/2)	*B*. *bovis* (1)	*Ehrlichia* sp. 7.2% (20/277)
		*Ehrlichia* sp. (1)	
		*T*. *annulata* (2)	
Şanlıurfa	*H*. *excavatum* (8/13)	*B*. *occultans* (1)	
		*Ehrlichia* sp. (6)	*Rickettsia* sp. 2.5% (7/277)
		*T*. *annulata* (5)	
	*Rh*. *turanicus* (2/3)	*Ehrlichia* sp. (1)	
		*T*. *annulata* (1)	
Karaman	*H*. *excavatum* (1/13)	*B. occultans* (1)	*T*. *annulata* 5.8% (16/277)
	*Rh*. *turanicus* (5/9)	*A*. *marginale* (1)	
		*Ehrlichia* sp. (2)	
		*Rickettsia* sp. (2)	

**Table 2 pathogens-11-00500-t002:** Pathogens identified in tick pools and GenBank accession numbers in this study.

DNA Sequences			High BLASTn Match				
Pathogen	Target Gene	Accession Number	Length (bp)	Query Cover (%)	Identity (%)	Reference Strains (Accession No.)	Source
*Anaplasma marginale*	*msp4*	OL408894	350	99	100	*A. marginale* (MF377455)	Cattle, Turkey
*Babesia bovis*	*sbp4*	OL408893	512	100	100	*B. bovis* (MN870661)	Buffalo, Egypt
*B. occultans*	18S rRNA (V4)	OL377855	403	100	100	*B. occultans* (KP745626)	Cattle, Turkey
*Coxiella* sp.	16S rRNA	OL413002	1,453	100	99.66	*Candidatus* Coxiella mudrowiae (CP011126)	*Rh. turanicus*, Israel
*Ehrlichia* sp.	*pCS20*	OL408895	279	100	99.28	Uncultured *Ehrlichia* sp. (KT362172)	*Rhip**icephalus microplus*, China
*Rickettsia* sp.	16S rRNA	OL377895	436	99	99.77	*R. aeschlimannii* (KY229715)	*Rh. sanguineus*, Lebanon
		OL377896	439	100	99.77	*Candidatus* Rickettsia barbariae (EU272189)	*Rh. turanicus*, Italy
		OM541405	366	98	100	*Rickettsia* sp. Chad (AF510102)	Human, Chad
		OM541406	436	98	99.29	*R. aeschlimannii* (KY229715)	*Rh. sanguineus*, Lebanon
		OM541407	439	99	99.77	*Candidatus* Rickettsia barbariae (EU272189)	*Rh. turanicus*, Italy
	*gltA*	OM717962	401	100	100	*Rickettsia* sp. Chad (AF510103)	Human, Chad
		OM717963	401	100	100	*R. aeschlimannii* (KU961540)	*H. marginatum*, Crimea
		OM717964	401	100	100	*Candidatus* Rickettsia barbariae (MF002503)	*Rh. turanicus*, China
	*ompA*	OM717959	210	97	99.05	*Rickettsia* sp. Chad (AH011786)	Human, Chad
		OM717960	210	98	99.52	*R. aeschlimannii* (MK922658)	*H. marginatum*, Turkey
		OM717961	210	99	100	*Candidatus* Rickettsia barbariae (MN482129)	*Rh. turanicus*, Turkey
*Theileria annulata*	*Tams-1*	OL408892	452	95	99.07	*T. annulata* (KX130956)	Dog, Tunisia

## Data Availability

The datasets generated during and/or analyzed during the current study are available from the corresponding author on reasonable request.

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
