# Peer review of "Protozoan and Rickettsial Pathogens in Ticks Collected from Infested Cattle from Turkey"

_pathogens, 2022, doi:10.3390/pathogens11050500_

Round 1

Reviewer 1 Report

Although the number of cattle (n=22), from which ticks were collected is small the tick-borne pathogens found in the study contribute to the literature. The article is well written, but there is no table or result on which region (Diyarbakir, Gaziantep, Kahramanmaras, Karaman, and Urfa) the pathogens were reported. In addition, no information was shared about why these regions were chosen. I believe that the study would have been more original if the authors had also taken blood samples from cattle from which tick samples were collected. In addition no information about health situation of cattle, and no information about how many ticks collected (species, number) each cattle (maximum-minumum number). My additional comments about the article are below.

Lines 49-51: To date, 702 species of Ixodidae (hard ticks), 193 species of Argasidae (soft ticks), and one species of Nuttalliellidae (Nuttalliella namaqua) have been described worldwide

Please check number of ticks (Ixodidae-731 and Argasidae-216) in recently published (Dantas-Torres et al. 2018).

Lines 75, 77, 250: 16S rDNA should be italic.

Lines 146-147: Understanding the complex relationships between ticks and pathogens is important to public health in mitigating tick-borne pathogen (TBP) transmission.

It’s not only public health… Please check that?

References: Please check protozoon name? They should be italic! (Lines 326,336, 352……….)

4.1. Ethical Statements

Do you have also the ethical statements for collection tick sample from cattle?

Line 170, 171: B. bovis 24.4%?? What kind of tick species pool? This result was obtained in ticks collected from which region? Please mention that!

-Why authors didn’t collect blood from cattle?

Line 201-201: It's nice that Ehrlichia ruminantum is reported for the first time, but from which region was it reported? In addition, If you collect same animal those all tick sample, its make sense 4 different species infected with E. ruminantum.

Fig 1: Could you please add number of cattle in the Fig.1?

In addition, authors should be calculate and add 95% CI information for positive samples.

Author Response

Reviewer 1

Comments and Suggestions for Authors

  1. Although the number of cattle (n=22), from which ticks were collected is small the tick-borne pathogens found in the study contribute to the literature. The article is well written, but there is no table or result on which region (Diyarbakir, Gaziantep, Kahramanmaras, Karaman, and Urfa) the pathogens were reported.

Response: We appreciate your comments. According to the reviewer's suggestion, we updated Figure 1 and added Table 1 showing the information based on region.

  1. In addition, no information was shared about why these regions were chosen. I believe that the study would have been more original if the authors had also taken blood samples from cattle from which tick samples were collected.

Response: Thank you for the comments. The provinces located in the south of Turkey have a warmer climate compared to the provinces in other regions, and the active periods of ticks during the year are longer in these provinces than in many provinces of Turkey. For this reason, tick samples were collected from Diyarbakir, Gaziantep, Kahramanmaras, Karaman, and Şanlıurfa provinces. In addition, the fact that some of these provinces are located in regions close to the border has been an important factor in their preference. We attempted to collect cattle blood samples, but the owners did not provide permission for collection.

  1. In addition, no information about health situation of cattle, and no information about how many ticks collected (species, number) each cattle (maximum-minumum number). My additional comments about the article are below.

Response: Tick samples were collected from apparently healthy cattle. In total, 277 ticks were collected from 22 cattle. Minimum and maximum number of ticks were 1 (from Diyarbakır) and 45 (from Kahramanmaraş), respectively. Line: 280-281.

  1. Lines 49-51:To date, 702 species of Ixodidae (hard ticks), 193 species of Argasidae (soft ticks), and one species of Nuttalliellidae (Nuttalliella namaqua) have been described worldwide. Please check number of ticks (Ixodidae-731 and Argasidae-216) in recently published (Dantas-Torres et al. 2018).

Response: We thank the reviewer for pointing it out. We have modified our sentences and updates the reference. Line: 48-50.

  1. Lines 75, 77, 250:16S rDNA should be italic.

Response: We have corrected it.

  1. Lines 146-147:Understanding the complex relationships between ticks and pathogens is important to public health in mitigating tick-borne pathogen (TBP)  It’s not only public health… Please check that?

Response: We have modified the sentence to “Understanding the complex relationships between ticks and pathogens is important to controlling tick-borne pathogen (TBP) transmission. Line: 146-148.

  1. References: Please check protozoon name? They should be italic! (Lines 326,336, 352……….)

Response: We have checked the references and corrected it throughout the manuscript.

  1. 1. Ethical Statements Do you have also the ethical statements for collection tick sample from cattle?

Response: We have added it in Line 269-270. “The owners of the cattle were informed about the study, and their approvals were obtained for tick collection.”

  1. Line 170, 171: bovis 24.4%?? What kind of tick species pool? This result was obtained in ticks collected from which region? Please mention that!

Response: We have modified Table 1-2 and showed detailed information about the detection of pathogens in the tick pools. We also modified the sentence in 170-172.

  1. -Why authors didn’t collect blood from cattle?

Response: Despite the owners agreeing to collect the ticks infesting cattle, unfortunately, we did not get the permission to collect blood samples from their animals.

  1. Line 201-201:It's nice that Ehrlichia ruminantum is reported for the first time, but from which region was it reported? In addition, If you collect same animal those all tick sample, its make sense 4 different species infected with  ruminantum.

Response: In Table 1 and Table 2, we indicated that the Ehrlichia sp. infection were detected in Gaziantep (4), KahramanmaraÅŸ (7), Åžanlıurfa (7), and Karaman (2). Moreover, we agree that there is a high possibility that infection with the same pathogens in different ticks collected from the same animal and that infection status of the host cattle could have complemented the data we obtained. However, this is one of the limitations of the present study.

  1. Fig 1:Could you please add number of cattle in the Fig.1?

Response: We have modified Figure 1 and add the number of cattle in each region.

  1. In addition, authors should be calculate and add95% CI information for positive samples.

Response: We added it in Table 1, wherein the descriptive statistics are shown.

Reviewer 2 Report

This manuscript reports PCR screening of Hyalomma and Rhipicephalus spp. ticks collected from cattle in Turkey for presence of DNA of two protozoan parasite genera, Babesia and Theileria, and three genera of intracellular tick-borne bacteria: Anaplasma, Ehrlichia and Rickettsia. Representatives of all five genera were detected and the authors carried out some phylogenetic analysis of the bacteria.

General comments:

The manuscript requires some improvement to bring it up to publication standard in terms of English (recommend using an English language editing service) and interpretation of the results.

A table summarising the tick collection data (numbers of ticks of each species collected in each province and dates of collection) should be included. If there was a requirement for any Turkish Government ethical approval and/or permission to collect ticks, this should be stated.

The references for all the PCRs used should be included in the text in section 4.4, not just the species-specific assays. In Table S1, the reference numbers should be included.

The authors should explain why they did not sequence the 16S rRNA amplicons obtained from Anaplasma- and Ehrlichia-positive samples; this data would strengthen their phylogenetic analyses. The authors should also be more careful in how they describe the results from the Coxiella and Ehrlichia PCRs: in section 2.2 they should say that positive PCR products (or amplicons) were obtained using the Coxiella and Ehrlichia PCRs, because their further analyses in sections 2.3 and 2.4 indicated that they did not detect either Coxiella burnetii or Ehrlichia ruminantium. All four tick species identified in this study are reported to carry Coxiella-like endosymbionts (doi: 10.1111/mec.14094; https://doi.org/10.1186/s13071-019-3862-4), which presumably would be detected by the PCR used. E. ruminantium is the causative agent of heartwater, a severe, often fatal disease of ruminants, and ticks become infected with E. ruminantium by feeding on an infected vertebrate host. It would be very surprising to find such high prevalence of E. ruminantium in ticks in Turkey in the complete absence of any reports of heartwater disease in cattle or small ruminants. The authors should therefore either reword the manuscript to clarify that the sequences they detected belonged to, respectively, Coxiella sp. and Ehrlichia sp. bacteria (as their results show, the closest matches were to bacteria of unknown pathogenicity: respectively, Candidatus Coxiella mudrowiae from Israel and an Ehrlichia sp. from China – not C. burnetii or E. ruminantium), or they should carry out amplification of additional genes such as groEl for Coxiella e.g. https://doi-org.liverpool.idm.oclc.org/10.1016/j.ttbdis.2014.04.003 and map1 for Ehrlichia e.g. https://doi.org/10.1007/s10493-007-9073-2 in support of their naming of the detected bacterial species.

The Discussion would be greatly strengthened by consideration of the origins of the protozoa and bacteria detected in the ticks removed from cattle – did they originate in the tick (resulting from either transstadial transmission, in the case of Theileria and Ehrlichia, or transovarial transmission, in the case of Rickettsia, Coxiella-like endosymbionts and potentially Babesia, or both) or in the bloodmeal obtained from the host on which they were feeding? The authors should also discuss the effect of having pooled the ticks on the detection of pathogens, particularly on prevalence of co-infections. It would be helpful to know the numbers of ticks in each pathogen-positive pool. It is not clear from Table 2 which single and co-infections belong to H. anatolicum and which to R. turanicus, possibly because R. turanicus is centred in its cell in the “Infected tick species” column, while the other three tick names are positioned at the top of their respective cells? This should be corrected.

All species names of ticks and pathogens should be given in full the first time they are mentioned in the abstract, in the text and in tables.

The References are not listed in the order in which they appear in the text, and very few of the species names are italicised. This should be corrected.  

Specific points:

Lines 23-24: change to “Tick-borne diseases are caused by many pathogens of public health and veterinary importance, including bacteria, viruses and protozoa. They are a barrier to global…”

Line 34: change to “…followed by Ehrlichia sp. (22.2%)…, Coxiella sp. (10%)…”

Line 86: change to “…Coxiella sp., Ehrlichia sp.,…”

Line 87: mention which genes were amplified from each pathogen

Table 2 and lines 150, 184, 195, 196 etc: change all mentions of C. burnetii and E. ruminantium detected in this study to Coxiella sp. and Ehrlichia sp.

Line 110 and 112: Table 2 does not present any data on sequence identities. Should this be Table 3?

Line 114: change to “The phylogenetic analysis of the sequences obtained from the PCRs specific for A. marginale, E. ruminantium and Rickettsia spp. was performed…

Line 118: change to “the Ehrlichia sp. pCS20 sequence…”

Line 128: which Rickettsia sp.?

Line 157 and 165: “major surface protein 4 (MSP4)” should not be italicised – it is a protein. The corresponding gene, “msp4” is italicised.

Line 183: “Boophilus annulatus” is now known as “Rhipicephalus annulatus”, as in line 173.

Line 214: Does “This study” refer to reference #47, or to the current manuscript? If the latter, change to “The present study”

Line 244: define “severe” and indicate where the cattle with severe infestations were located (i.e., were they all at the same sample site, or in more than one province?), either here or in a table summarising the tick collections (see general comment above).

In Table 3, what is the difference between the sequences OL377895 and OM541406, and between OL377896 and OM541407? Were they obtained from different provinces, or from different tick species? This should be clarified and mentioned in the text.

Author Response

Reviewer 2

This manuscript reports PCR screening of Hyalomma and Rhipicephalus spp. ticks collected from cattle in Turkey for presence of DNA of two protozoan parasite genera, Babesia and Theileria, and three genera of intracellular tick-borne bacteria: AnaplasmaEhrlichia, and Rickettsia. Representatives of all five genera were detected and the authors carried out some phylogenetic analysis of the bacteria.

General comments:

  1. The manuscript requires some improvement to bring it up to publication standard in terms of English (recommend using an English language editing service) and interpretation of the results.

Response: We have thoroughly checked the grammar and spelling throughout the manuscript. In addition, we have rewritten some parts of the results and discussion section for more clarity and brevity.

  1. A table summarising the tick collection data (numbers of ticks of each species collected in each province and dates of collection) should be included. If there was a requirement for any Turkish Government ethical approval and/or permission to collect ticks, this should be stated.

Response: We have modified Figure 1 and included some information about the ticks in Table 2. Regarding the reviewer’s suggestion, we revised our ethical approval statement in Line: 269-270 and we added the sample collection date in section 4.2. Line: 275.

  1. The references for all the PCRs used should be included in the text in section 4.4, not just the species-specific assays. In Table S1, the reference numbers should be included.

Response: Thank you for pointing these mistakes. We cited all the PCR assays used in the study accordingly. In addition, we corrected the citation errors in Table S2.

  1. The authors should explain why they did not sequence the 16S rRNA amplicons obtained from Anaplasma- and Ehrlichia-positive samples; this data would strengthen their phylogenetic analyses.

Response: We thank the reviewer for the comment. The initial objective of using the PCR targeting 16S rRNA of Anaplasma/Ehrlichia was to identify samples positive for Anaplasma/Ehrlichia. Then, we amplified the major Anaplasma/Ehrlichia agents of TBDs in cattle (A. marginale and E. ruminantium). After, we confirmed that all 16S rRNA-positive samples were also positive with either A. marginale and Ehrlichia sp. We have added this statement for clarity in section 2.2. Line: 94-96.“In the case of Anaplasma/Ehrlichia, all samples that showed an amplicon using the 16S rRNA assay were positive using A. marginale MSP4 and/or E. ruminantium pCS20 assays, thus, amplicons from the latter assays only were sequenced.”

  1. The authors should also be more careful in how they describe the results from the Coxiellaand Ehrlichia PCRs: in section 2.2 they should say that positive PCR products (or amplicons) were obtained using the Coxiella and Ehrlichia PCRs, because their further analyses in sections 2.3 and 2.4 indicated that they did not detect either Coxiella burnetii or Ehrlichia ruminantium.

Response: We have modified the naming of Coxiella and Ehrlichia in section 2.2.

  1. All four tick species identified in this study are reported to carry Coxiella-like endosymbionts (doi: 10.1111/mec.14094; https://doi.org/10.1186/s13071-019-3862-4), which presumably would be detected by the PCR used. ruminantium is the causative agent of heartwater, a severe, often fatal disease of ruminants, and ticks become infected with E. ruminantium by feeding on an infected vertebrate host. It would be very surprising to find such high prevalence of E. ruminantium in ticks in Turkey in the complete absence of any reports of heartwater disease in cattle or small ruminants. The authors should therefore either reword the manuscript to clarify that the sequences they detected belonged to, respectively, Coxiella sp. and Ehrlichia sp. bacteria (as their results show, the closest matches were to bacteria of unknown pathogenicity: respectively, Candidatus Coxiella mudrowiae from Israel and an Ehrlichia sp. from China – not C. burnetii or E. ruminantium), or they should carry out amplification of additional genes such as groEl for Coxiella e.g. https://doi-org.liverpool.idm.oclc.org/10.1016/j.ttbdis.2014.04.003 and map1 for Ehrlichia e.g. https://doi.org/10.1007/s10493-007-9073-2 in support of their naming of the detected bacterial species.

Response: Thank you for the suggestion. We have incorporated a paragraph on the detection of Coxiella and Ehrlichia endosymbionts in the discussion section. Line 228-236.“The detection of Coxiella-like and Ehrlichia-like species from ticks in this study despite using specific PCR primers, coupled with the results in BLASTn and phylogenetic analysis, may indicate that these species are endosymbionts. Endosymbionts are non-pathogenic microorganisms that have evolved in the carriers or hosts either as obligate or facultative symbionts [36]. Ticks were previously reported to be highly infected with endosymbionts [37,38]. In this study, an isolate highly similar to Candidatus Coxiella mudrowiae, a known Coxiella endosymbiont, was detected in Rh. turanicus ticks, as it was similarly described in a previous study [39]. As to what extent the endosymbionts of ticks in Turkey affect transmission of other TBPs needs to be studied in the future.

  1. The Discussion would be greatly strengthened by consideration of the origins of the protozoa and bacteria detected in the ticks removed from cattle – did they originate in the tick (resulting from either transstadial transmission, in the case of Theileria and Ehrlichia, or transovarial transmission, in the case of RickettsiaCoxiella-like endosymbionts and potentially Babesia, or both) or in the bloodmeal obtained from the host on which they were feeding? The authors should also discuss the effect of having pooled the ticks on the detection of pathogens, particularly on prevalence of co-infections. It would be helpful to know the numbers of ticks in each pathogen-positive pool. It is not clear from Table 2 which single and co-infections belong to  anatolicum and which to R. turanicus, possibly because R. turanicus is centred in its cell in the “Infected tick species” column, while the other three tick names are positioned at the top of their respective cells? This should be corrected.

Response: The paragraph below was added to the discussion section for clarity. Line: 237-251. “In this study, a large number of single or multiple pathogenic microorganisms’ DNA was detected in cattle tick samples. It is difficult to determine whether these microorganisms are caused by ticks or by hosts since all ticks were collected while sucking blood from cattle. Karim et al. [37] reported that the characterization of multiple infections in ticks poses a major scientific challenge for understanding the epidemiology of tick-borne infectious diseases. It is thought that the high prevalence in tick pools may be related to the number of ticks forming the pools in this study. It was expected that the infection rate would be lower in pools with a low number of ticks, and a higher infection rate in pools with a high number of ticks. The fact that the tick DNA samples were not extracted from each tick individually but from 1-10 ticks collected in the pools limits the discussion of single and coinfection status. In the present study, in addition to single infections (30/52), several dual (15/52), triple (5/52), and quadruple (2/52) coinfections were detected. However, our study does not discuss the co-transmission of many microorganisms from ticks to cattle, and further studies are needed to obtain information about potential tick-borne pathogens and the dynamics of tick-borne infections in a region [37,38].”

  1. All species names of ticks and pathogens should be given in full the first time they are mentioned in the abstract, in the text and in tables.

Response: We have checked and modified it.

  1. The References are not listed in the order in which they appear in the text, and very few of the species names are italicised. This should be corrected.

Response: We thank reviewer for pointing it out and we have corrected these mistakes.

Specific points:

  1. Lines 23-24: change to “Tick-borne diseases are caused by many pathogens of public health and veterinary importance, including bacteria, viruses and protozoa. They are a barrier to global…”

Response: We thank the reviewer’s suggestion and we have changed the sentence. Line: 23-25.

  1. Line 34: change to “…followed by Ehrlichia (22.2%)…, Coxiellasp. (10%)…”

Response: We have changed as the reviewer had suggested. Line: 33-34.

  1. Line 86: change to “…CoxiellaEhrlichiasp.…”

Response: We have changed correspondingly. Line: 82.

  1. Line 87: mention which genes were amplified from each pathogen

Response: As the reviewer had suggested, we have corrected our description to “The target genes of species-specific PCR were A. marginale major surface protein 4 (msp4), B. bovis spherical body protein 4 (sbp4), B. occultans 18S rRNA V4 hypervariable region, Coxiella sp. 16S rRNA, Ehrlichia sp. pCS20, Rickettsia sp. (16S rRNA, gltA and ompA), and T. annulata merozoite surface antigen 1 (Tams-1).Line:83-87.

  1. Table 2 and lines 150, 184, 195, 196 etc: change all mentions of  burnetii and E. ruminantiumdetected in this study toCoxiella sp. and Ehrlichia sp.

Response: We have changed it to the reviewer’s suggestion.

  1. Line 110 and 112: Table 2 does not present any data on sequence identities. Should this be Table 3

Response: We thank reviewer for pointing out our mistakes and we have corrected it.

  1. Line 114: change to “The phylogenetic analysis of the sequences obtained from the PCRs specific for  marginaleE. ruminantiumand Rickettsia spp. was performed…

Response: We have changed it to the reviewer’s suggestion. Line: 327-328.

  1. Line 118: change to “the Ehrlichia pCS20 sequence…”

Response: We have modified it as the reviewer had suggested. Line: 118.

  1. Line 128: which Rickettsia sp.?

Response: We added the specific species “Rickettsia sp. Chad”. Line: 128.

  1. Line 157 and 165: “major surface protein 4 (MSP4)” should not be italicised – it is a protein. The corresponding gene, “msp4” is italicised.

Response: We have modified it. Line:157, 159 and 164.

  1. Line 183: “Boophilus annulatus” is now known as “Rhipicephalus annulatus”, as in line 173.

Response: We have changed “Boophilus annulatus” to “Rh. annulatus”

  1. Line 214: Does “This study” refer to reference #47, or to the current manuscript? If the latter, change to “The present study”

Response: “This study” indicated the current manuscript, and we have change to “The present study……”. Line: 209.

  1. Line 244: define “severe” and indicate where the cattle with severe infestations were located (i.e., were they all at the same sample site, or in more than one province?), either here or in a table summarising the tick collections (see general comment above).

Response: The word “severe” used here is subjectively written. Tick infestation status may be interpreted differently by other investigators. In the study, the number of ticks collected from Diyarbakir province is quite low (2 ticks from 2 cattle), and in fact, it is wrong to describe the infestations here as severe. However, the average number of ticks was determined as 8 in Karaman, 16 in Gaziantep, 17 in Åžanlıurfa, and 22 in KahramanmaraÅŸ, and tick infestation was subjectively described here as severe. We have added information in Figure 1.

  1. In Table 3, what is the difference between the sequences OL377895 and OM541406, and between OL377896 and OM541407? Were they obtained from different provinces, or from different tick species? This should be clarified and mentioned in the text.

Response: We thank the reviewer for the question. We have mentioned in Line 108-111. “OL377895 and OM541406 were detected in different H. excavatum pools which were collected from different cattle in KahramanmaraÅŸ. Meanwhile, OL377896 was detected in H. excavatum in KahramanmaraÅŸ, while OM541407 was detected in Rh. turanicus in Karaman.”

Reviewer 3 Report

The present paper represents an attempt in characterizing the distribution and prevalence of tick-borne pathogens (TBPs) in few areas of Turkey. Despite the extensive molecular lab work done, there are several flaws and elements which needs to be corrected before publication can occurs.

Some important and general points which are needed to be fully addressed by the authors:

  • There are several points where the English is not good, making the sentences often ambiguous or not clear I strongly advise the authors to contact a native speaker English to do a final check on the manuscript which the authors will provide, after the corrections.
  • In all your PCR analysis report, you just specify the identity percentage of the blasted sequence, which, of course, is an important feature to report. Nevertheless, it is also important to specify the query cover (i.e., how your query cover was long). I assume that also the query cover percentage will be 100% but I want just to doublecheck. I’m being very pedantic on this step because since all these TBPs (especially Babesias but not only) are genetically very similar and if you blast a sequence that is too short, it might give you 100% ident to more than one pathogen species. For this reason, it should be added in multiple parts of the manuscript.
  • As a matter of style, I think that some sentences reported in the discussion (as in lines 157-159) where all the genes used for characterization of TBPs (i.e. MSP4, gltA) are introduced might be moved to the introduction, to get the authors to familiarize with such terms before seeing numerous unannounced acronyms in the results section.

More in details:

Abstract

Lines 23-24: tick borne disease are NOT involved in important pathogens, probably it would be more correct to say the other way around. Furthermore, you end the sentences saying that they include bacteria, viruses and protozoa, but what about Rickettsial organism (they do not fully fall into the bacteria reign). Additionally “barrier on global animal production” doesn’t make sense, as it is.
I strongly suggest rephrasing it correctly.

Lines 25-26: remove “and update of information”

Line 26: contribute to improving (correct or not?)

Line 30: Add s to both “male and female”. Plus, add tick before “species”.

Line 31: I would rephrase “Tick species were identified first morphologically, then confirmed via PCR” or something similar.

Line 32: replace “method” with assay or just remove it.

Line 33: Since you used pool of ticks for PCR, I don’t know whether the term “detection rate” would be suitable in this case. Please, double check whether this is still suitable in this case or substitute with something more appropriate (prevalence?).

Line 37: add the query cover range

Line 38: “Astrakhan fever Rickettsia” is a case from a human?

Lines 40-41: I think that the sentence “shall contribute to prevention and control of tickborne diseases in Turkey” is a little to heavy to use considering you just Molecularly identified TBPs in 270 ticks sampled from just 70 cattle in five regions; also, because there are no mentions in the paper on how the identification of TBPs might help in such prevention. I would rephrase as “it expands the knowledge of the epidemiology of TBPs in Turkey” or something similar, but not mentioning the prevention and control.

Introduction

Line 48: “world-wide distributed” sound very odd. Consider rephrasing.

Lines 55-56: I found the sentence “the rate of tick-infested animals is usually more than 20% of the herd” very confusing. You cannot understand which animals you are referring to (i.e. cattle? Wild ruminants?). Rephrase it.

Line 57: “important economy” doesn’t make sense.  Rephrase it.

Line 59 “Parasitic”. They are not just parasitic (also bacterial, etc.). Rephrase it.

Lines 61-62. “The main TBD suffered by cattle include etc. etc.” Remove the whole sentence, as it doesn’t add anything to the paper.

Line 63: “molecular surveillance of ticks and their harbored pathogens”. While molecular survey is a term more than acceptable to TBPs, it is not appropriate for ticks (as you can take survey them firstly macroscopically). Rephrase it.

Results

Lines 70-71: It is not clear what the number in brackets stands after the geographical location. Are they referring to animals? Or the number of farms infested?   Please correct.

Lines 74-77: the entire sequence is reporting just how things were performed, so it must go to the M&M section.

Line 80 and table 1: add query cover percentages in both line and table (suggestion: if the query cover is 100%, you can omit it from the table and just mentioning it in the table caption.

Lines 91 and 92: You mentioned that the two positive samples for Babesia occultans were confirmed by sequencing. Why just these two? What about the others?

Line 97: Consider use another word instead of “speciation”.

Lines 99-101: All these lines have to go to M&M.

Line 101: add some words (e.g. “our”) to clearly specify that the sequence which you detected showed “100 % identity etc.”.

Line 102: once again, add the query cover. You should do it the same for the remainder of the paper, although not precisely reported from here to the end.

Line 104: here you mention a “Coxiella-like pathogen”, which is not reported elsewhere. Why such a thing?

Line 114-115: All these lines have to go to M&M.

Lines 163-164: “The information of A marginale etc.” sounds very odd in English. Consider rephrasing it.

Lines 188-189: I found the historical note on the first human case of ehrlichiosis a little superfluous. Delete the entire sentence.

Line 193: replace the words “gene segments” with something else. I found it a very odd term.

Line 197: Reword the sentence because it isn’t clear which Ehrlichia you’re referring to (your isolate?). Similar modifications should be done all across the paper.

Lines 203-204: This conclusion is too heavy and, in my opinion, wrong. If a percentage identity (and without a query cover percentage) of 99.28% is enough to call another strain, we probably could find uncountable strains of parasites all days. Strains distinction must be not done only considering DNA sequence of a single gene but also using multiple genes, pathogenicity and morphology, at least and should be supported by also multiple other studies.  Consider also that slight genetic drift and possible sequencing puntiform errors might account for such little discrepancy. I suggest its deletion.

Line 225: after “Turkey” there is lack of a reference. Add accordingly

Line 232: Before going into conclusion, a think a paragraph discussing the limits of the study is needed at least citing the following two points. The first one is the small sample  of population: you just (I totally appreciate the intensity and the heavy lab work done here) evaluate multiple ticks coming from 22 cattle. This is not a high number and therefore you must interpret your work in light of the a restricted number of animals examined.
The second point is how you extracted the DNA from the ticks (see also further comments down below): of course the tick can or cannot be filled by recently sucked blood from the cow, and this can bias, of course the interpretation of the molecular results. Such point is pivotal to mention in my opinion.

M&M:

 Line 244: mention how the cattles were “examined in terms of tick infestation and also can you give why they were considered severe infested? It would be also interesting in which time of the year they were collected.

Line 247: Did all ticks examined were attached to the cattle bloodfeeding? if so, when they were removed, some prt of the capitulum remained attached to the skin of the cattle? If so, can this also have impacted on the recognition of the tick species

Lines 252-253: it is not clear enough how did you decided how to do the pooling for the tick DNA extraction. 

4.4. It isn't clear enough how did you the PCR and the scheme you attained. Did you first do a PCR usings the mitochondrial 16S or 18S? and then ,on just the positive, you perfomred the species specific PCR (e.g. MSP4) ?  reword it and make it more clear.

4.6. This part need to be expanded. Why did you used the maximum likelihood? and how did you pick the other isolates for comparison? 

Author Response

Review 3.

Comments and Suggestions for Authors

The present paper represents an attempt in characterizing the distribution and prevalence of tick-borne pathogens (TBPs) in few areas of Turkey. Despite the extensive molecular lab work done, there are several flaws and elements which needs to be corrected before publication can occurs. Some important and general points which are needed to be fully addressed by the authors:

  1. There are several points where the English is not good, making the sentences often ambiguous or not clear I strongly advise the authors to contact a native speaker English to do a final check on the manuscript which the authors will provide, after the corrections.

Response: We appreciate your comments. We have thoroughly checked the grammar and spelling throughout the manuscript. In addition, we have rewritten some parts of the results and discussion section for more clarity and brevity.

  1. In all your PCR analysis report, you just specify the identity percentage of the blasted sequence, which, of course, is an important feature to report. Nevertheless, it is also important to specify the query cover (i.e., how your query cover was long). I assume that also the query cover percentage will be 100% but I want just to doublecheck. I’m being very pedantic on this step because since all these TBPs (especially Babesias but not only) are genetically very similar and if you blast a sequence that is too short, it might give you 100% ident to more than one pathogen species. For this reason, it should be added in multiple parts of the manuscript.

Response: We agree with reviewer and we have added the query cover (%) in Table 3 and S1.

  1. As a matter of style, I think that some sentences reported in the discussion (as in lines 157-159) where all the genes used for characterization of TBPs (i.e. MSP4, gltA) are introduced might be moved to the introduction, to get the authors to familiarize with such terms before seeing numerous unannounced acronyms in the results section.

Response: We have mentioned the full names of the genes detected prior to enumerating the results in section 2.2.

More in details:

Abstract

  1. Lines 23-24: tick borne disease are NOT involved in important pathogens, probably it would be more correct to say the other way around. Furthermore, you end the sentences saying that they include bacteria, viruses and protozoa, but what about Rickettsial organism (they do not fully fall into the bacteria reign). Additionally “barrier on global animal production” doesn’t make sense, as it is. I strongly suggest rephrasing it correctly.

Response: The sentences “Tick-borne diseases are involved in many public health and veterinary important pathogens, including bacteria, viruses, and protozoa. It is a barrier on global animal production, especially in tropical and subtropical regions, including Turkey.” were changed as follows: “Diseases caused by tick-transmitted pathogens lick bacteria, virues, parasite, and rickettsia adversely affect both veterinary and human medicine, especially in tropical and subtropical regions, including Turkey.” Line 23-25.

  1. Lines 25-26: remove “and update of information”

Response: We have removed it.

  1. Line 26: contribute to improving (correct or not?)

Response: We have corrected it. Line: 25

  1. Line 30: Add s to both “male and female”. Plus, add tick before “species”.

Response: We have edited both of them. Line: 29.

  1. Line 31: I would rephrase “Tick species were identified first morphologically, then confirmed via PCR” or something similar.

Response: We have replaced the sentence to “A total of 277 adult ticks (males and females) were collected. After microscopic identification, tick pools were generated according to tick species, host animal, and sampling site for DNA extraction.” Line: 28-30.

  1. Line 32: replace “method” with assay or just remove it.

Response: We have removed it.

  1. Line 33: Since you used pool of ticks for PCR, I don’t know whether the term “detection rate” would be suitable in this case. Please, double check whether this is still suitable in this case or substitute with something more appropriate (prevalence?).

Response: We thank the reviewer for your helpful comment. We have checked some paper and changed the term to “prevalence”.

  1. Line 37: add the query cover range

Response: We have added the query cover range in Line: 36.

  1. Line 38: “Astrakhan fever Rickettsia” is a case from a human?

Response: We appreciate your comments. As the reference reported, a human clinical case of Astrakhan fever was reported in Chad before. Line: 213-214. (Reference: https://doi.org/10.1111/j.1749-6632.2003.tb07356.x.).

  1. Lines 40-41: I think that the sentence “shall contribute to prevention and control of tickborne diseases in Turkey” is a little too heavy to use considering you just Molecularly identified TBPs in 270 ticks sampled from just 70 cattle in five regions; also, because there are no mentions in the paper on how the identification of TBPs might help in such prevention. I would rephrase as “it expands the knowledge of the epidemiology of TBPs in Turkey” or something similar, but not mentioning the prevention and control.

Response: As suggested by the reviewer, we have edited it. Line: 39-40.

Introduction

  1. Line 48: “world-wide distributed” sound very odd. Consider rephrasing.

Response: We have change to “The global distribution of …….” Line: 47.

  1. Lines 55-56: I found the sentence “the rate of tick-infested animals is usually more than 20% of the herd” very confusing. You cannot understand which animals you are referring to (i.e. cattle? Wild ruminants?). Rephrase it.

Response: We have deleted the confusing sentence “the rate of tick-infested animals is usually more than 20% of the herd”.

  1. Line 57: “important economy” doesn’t make sense.  Rephrase it.

Response: We have deleted “the cattle industry still remains an important economy and”.

  1. Line 59 “Parasitic”. They are not just parasitic (also bacterial, etc.). Rephrase it.

Response: We replaced “parasitic diseases arising from ticks…...” to “TBDs had a significant ……” Line: 56.

  1. Lines 61-62. “The main TBD suffered by cattle include etc. etc.” Remove the whole sentence, as it doesn’t add anything to the paper.

Response: We have removed the whole sentence as suggested by the reviewer.

  1. Line 63: “molecular surveillance of ticks and their harbored pathogens”. While molecular survey is a term more than acceptable to TBPs, it is not appropriate for ticks (as you can take survey them firstly macroscopically). Rephrase it.

Response: We have replaced the “molecular surveillance of ticks and their harbored pathogens” to “there is still limited data about pathogens carried or transmitted by ticks.” Line: 58.

Results

  1. Lines 70-71: It is not clear what the number in brackets stands after the geographical location. Are they referring to animals? Or the number of farms infested?   Please correct.

Response: We have corrected it. Line: 64.

  1. Lines 74-77: the entire sequence is reporting just how things were performed, so it must go to the M&M section.

Response: We have corrected it as the reviewer’s suggestion. Line:305.

  1. Line 80 and table 1: add query cover percentages in both line and table (suggestion: if the query cover is 100%, you can omit it from the table and just mentioning it in the table caption.

Response: We have added the query cover in Line: 72 and Table S1.

  1. Lines 91 and 92: You mentioned that the two positive samples for Babesia occultans were confirmed by sequencing. Why just these two? What about the others?

Response: For Babesia spp., firstly, we performed the primary screening by using 18S rRNA (V4), then we use specific primers (B. bovis sbp4 and B. bigemina RAP1a) to confirm the positive samples. Among the positive samples, two of them could not detected by specific primers. So, we confirmed these two samples by sequencing.

  1. Line 97: Consider use another word instead of “speciation”.

Response: We have changed the title to “Types of infection detected in tick pools based on regions”. Line: 97.

  1. Lines 99-101: All these lines have to go to M&M.

Response: We have deleted it. Because we have similar description in M&M. Line: 306-311.

  1. Line 101: add some words (e.g. “our”) to clearly specify that the sequence which you detected showed “100 % identity etc.”.

Response: We have modified it as “The NCBI BLASTn analysis of A. marginale, B. bovis, and B. occultanssequences obtained from this study showed 100% identity with the reference sequences……” Line: 101.

  1. Line 102: once again, add the query cover. You should do it the same for the remainder of the paper, although not precisely reported from here to the end.

Response: We have added the query cover throughout the manuscript. Line: 112.

  1. Line 104: here you mention a “Coxiella-like pathogen”, which is not reported elsewhere. Why such a thing?

Response: We discussed the relevance of detecting Coxiella-like species in the discussion section Line 228-236.

  1. Line 114-115: All these lines have to go to M&M.

Response: We have modified it as the reviewer’s suggestion. Line 327-329.

  1. Lines 163-164: “The information of A marginale etc.” sounds very odd in English. Consider rephrasing it.

Response: We have removed “the information of” and changed to “A. marginale infection in ticks were rarely…...”. Line: 163.

  1. Lines 188-189: I found the historical note on the first human case of ehrlichiosis a little superfluous. Delete the entire sentence.

Response: We have deleted the sentence as suggested by the reviewer.

  1. Line 193: replace the words “gene segments” with something else. I found it a very odd term.

Response: We have change “gene segments” to “partial sequence”. Line: 191.

  1. Line 197: Reword the sentence because it isn’t clear which Ehrlichia you’re referring to (your isolate?). Similar modifications should be done all across the paper.

Response: We have redescribed our result in Line: 195-197. “Although Ehrlichia sp. was detected in H. excavatum and Rh. bursa in studies conducted in Turkey, that Ehrlichia sp. sequence found was similar to E. canis and Ehrlichia sp. Omatjenne strain”.

  1. Lines 203-204: This conclusion is too heavy and, in my opinion, wrong. If a percentage identity (and without a query cover percentage) of 99.28% is enough to call another strain, we probably could find uncountable strains of parasites all days. Strains distinction must be not done only considering DNA sequence of a single gene but also using multiple genes, pathogenicity and morphology, at least and should be supported by also multiple other studies.  Consider also that slight genetic drift and possible sequencing puntiform errors might account for such little discrepancy. I suggest its deletion.

Response: We appreciate the reviewer's suggestion and removed the sentence.

  1. Line 225: after “Turkey” there is lack of a reference. Add accordingly

Response: We thank the reviewer for pointing it out. We have added a reference. Line: 221.

  1. Line 232: Before going into conclusion, a think a paragraph discussing the limits of the study is needed at least citing the following two points. The first one is the small sample of population: you just (I totally appreciate the intensity and the heavy lab work done here) evaluate multiple ticks coming from 22 cattle. This is not a high number and therefore you must interpret your work in light of the a restricted number of animals examined. 
    The second point is how you extracted the DNA from the ticks (see also further comments down below): of course the tick can or cannot be filled by recently sucked blood from the cow, and this can bias, of course the interpretation of the molecular results. Such point is pivotal to mention in my opinion.

Response: Thank you for the suggestion. We have incorporated a paragraph extensively detailing the limitation of the study. Line : 252-260.“The limitation of this study was that it was carried out on a restricted number of cattle. The collection of ticks at a certain time of the year and the lack of periodic tick collection according to the seasons are considered as other limiting factors of this study. The study is also limited in choosing sampling locations and, thus, could not cover the whole of Turkey to provide a better representation of tick and tick-borne pathogens. Another limiting factor is the inability to obtain blood from infested cattle. Although we know the importance of this, since the samples were collected from cattle grazing freely in the field, sufficient restraint could not be ensured and serious problems were encountered in the blood collection process.”

M&M:

  1. Line 244: mention how the cattles were “examined in terms of tick infestation and also can you give why they were considered severe infested? It would be also interesting in which time of the year they were collected.

Response: After the cattle were properly restrained, the undertail, perianal and inguinal regions, ears, eye circles, neck, and inner parts of the legs were checked by inspection and palpation in terms of tick attachment between April and June. It is expressed as “severe” because a large number of ticks were collected from each cattle (min 1- max 45), especially in Gaziantep, KahramanmaraÅŸ, and Åžanlıurfa, except Diyarbakır.

  1. Line 247: Did all ticks examined were attached to the cattle bloodfeeding? if so, when they were removed, some prt of the capitulum remained attached to the skin of the cattle? If so, can this also have impacted on the recognition of the tick species

Response: All ticks were removed during blood-feeding on cattle. The tick removal process was performed with curved-tipped forceps without damaging the capitulum. Therefore, no problems originating from the capitulum were encountered during the morphological of ticks.

  1. Lines 252-253: it is not clear enough how did you decided how to do the pooling for the tick DNA extraction. 

Response: We have redescribed the method of pooling. Line: 286-287.

  1. 4. It isn't clear enough how did you the PCR and the scheme you attained. Did you first do a PCR usings the mitochondrial 16S or 18S? and then ,on just the positive, you perfomred the species specific PCR (e.g. MSP4) ?  reword it and make it more clear.

Response: We have modified 4.4 and added some words or sentences. “All tick pool samples were primary screened…….” and “The positive samples were selected for further species-specific detection. Line: 305-310.

  1. 6. This part need to be expanded. Why did you used the maximum likelihood? and how did you pick the other isolates for comparison? 

Response: We included details on the method used for constructing the tree and choosing the isolates in section 4.6. “The maximum likelihood was employed as the method for tree construction because of its use of complex models to simulate biological reality and infer sequence evolution [56]. The sequences included in the phylogenetic analysis were chosen based on the BLASTn search results and geographical origin.”

Round 2

Reviewer 2 Report

The authors have made a good effort to address most of the points raised by the reviewers. However, in this reviewer’s opinion, several points still require improvement to strengthen and bring the manuscript up to publication standard. This reviewer apologises for not raising some of these points during the first review round.

The issue of detection rate/prevalence can be addressed most usefully by determining the minimum detection rate for each microorganism in each tick species. This overcomes the difficulty in determining detection rate or prevalence when using pooled tick samples. Once calculated, this information should be presented in a Table and throughout the Abstract and text to replace “infection rate” or "prevalence". Calculating the minimum detection rate for a microorganism/tick species combination is done as follows, using the example of the combination of Theileria annulata and Hyalomma excavatum. A total of 203 H. excavatum ticks were collected (line 68); from Table 2, the number of T. annulata-positive H. excavatum pools is 11. This means that a minimum of one tick in each of the 11 pools was positive for T. annulata (regardless of the number of ticks in the pool, and there is no way of knowing whether more than one tick was positive in any single pool). The minimum detection rate for T. annulata in H. excavatum is therefore [11 x 100] ÷ 203 = 5.4%. 

In this reviewer’s opinion, the new Table 1 does not serve any useful purpose, as Table 2 has also been revised to present the results of microorganism detection by province. Although another reviewer asked for confidence limits and odds ratios to be included, the purpose of that information is not clear to this reviewer and it does not add anything to the interpretation of the data (the study is not about the effectiveness of the detection methods used, nor a comparison between different provinces or tick species; it is a descriptive study).

The term “co-infection” should only be applied to samples containing DNA from a single tick. If there is more than one tick in a pool, there is no way of knowing which detected microorganisms were in each of those ticks, and whether any tick was carrying more than one microorganism. Therefore, and bearing in mind the points raised above about minimum detection rate, this reviewer suggest that the new Table 1 is discarded, and Table 2 is amended to contain columns labelled “Province”; “Tick Species (Positive pools/Total pools)”; “Microorganism Detected (No. of pools)”; “Minimum Detection Rate”. The first row would then read (from left to right) “Gaziantep”; Hyalomma excavatum (8/X)”; “Theileria annulata (4)”; “5.4%” (where X is the total number of H. excavatum pools from Gaziantep). The second row would contain the equivalent data for Babesia bovis detected in pools of H. excavatum from Gaziantep, and so on.

If any pool containing only one tick was positive for more than one microorganism, this can be reported in the text as a co-infection, and discussed as such.

In Figure 1, the number of R. turanicus ticks collected in Sanhurfa is given as three, but in Table 2 there are four positive R. turanicus pools (each presumably containing at least one tick!), including one pool positive for both Ehrlichia sp. and T. annulata. This should be checked and corrected (either there should be more ticks or fewer pools).

This reviewer questions whether or not one tick per cow can be considered as a "severe" infestation (authors' response to reviewer #3).

Abstract, text and Tables 2 and 3 – ALL species names should be in full the first time they are mentioned in all of these locations.

This reviewer strongly recommends that the authors screen the Ehrlichia-positive DNA samples with an Ehrlichia-specific 16S rRNA PCR and sequence at least one PCR product from each tick species. In Genbank there are 26 times more 16S rRNA sequences than pCS20 sequences deposited for Ehrlichia spp., so this gene should provide more information about the species present in the Turkish ticks. Although it is unlikely that E. ruminantium is present in Turkey, it is important from an animal health perspective to exclude it and, if possible, identify what Ehrlichia species is/are present and if they pose a risk to livestock.  

Lines 228-230: the Ehrlichia sp. is very unlikely to be an endosymbiont. As far as this reviewer knows, there are no reports of Ehrlichia spp. behaving as symbionts, and they are not normally transovarially transmitted (except possibly in the case of the one-host tick R. microplus, see DOI 10.1186/s13071-016-1651-x). Ticks acquire Ehrlichia by feeding on an Ehrlichia-infected host.

Tables S1 and S2 – ALL species names should be in full, so in Table S1 Hyalomma excavatum, Hyalomma anatolicum and Rhipicephalus bursa should all be in full, and in Table S2 Theileria ovis, Theileria annulata, Babesia bovis etc should all be in full. In Table S2, there should be an indication of which primer is the forward and which is the reverse for each pair, either in front of each primer, or by stating in the Table title that the forward primer is listed first in each case. Table S2 title should be change to “Primer sets used…”

Specific points:

Lines 23-24: change to “Diseases caused by tick-transmitted pathogens including bacteria, viruses and protozoa are of veterinary and medical importance especially in…”. Diseases do not adversely affect medicine, they adversely affect the humans and animals suffering from disease. Rickettsia are alphaproteobacteria belonging in the domain “Bacteria”; there is no need to list them separately from bacteria.

Lines 33-34: change prevalences to minimum detection rates as explained in general comments below, and using the figures calculated in the specific comment on lines 87-90 below.

Line 36: change to “All sequences obtained from…”

Line 39: change to “and expand the knowledge”

Lines 52-53: change to “(43 from the family Ixodidae and 8 from the family Argasidae).”

Line 54: delete “(Boophilus)” – otherwise your statement excludes ticks such as R. bursa, R. turanicus and R. sanguineus.

Line 75: change to “Figure 1. Map of sample collection sites in Turkey showing numbers of cattle examined (no. infested/total) and numbers of ticks of each species collected.

Lines 87-90: change to: “The most frequently-detected pathogens in this study (Table 2) were B. bovis (minimum detection rate 7.9%, 22 positive pools), followed by Ehrlichia sp. (minimum detection rate 7.2%,20 positive pools), T. annulata (minimum detection rate 5.8%, 16 positive pools), Coxiella sp. (minimum detection rate 3.3%, 9 positive pools), A. marginale and Rickettsia spp. (minimum detection rate 2.5%, 7 positive pools each) and B. occultans (minimum detection rate 0.7%, 2 positive pools).

Lines 103-105: which tick species was the sequenced sample obtained from? This should be stated. Also change to “The Coxiella-like sequence (OL413002)…” – there is no evidence from your study, or from the original article (doi:10.1093/gbe/evv108) that this Coxiella is a pathogen, i.e. a disease-causing agent.

Line 150: change to “Overall, seven tick-associated microorganisms, including…” – it is not known whether or not the Coxiella sp. is transmitted by ticks to hosts during feeding.

Line 165: change to “was highly similar to”

Lines 181-188: insert a statement saying that more work is needed to determine whether or not any of the tick species sampled in the present study were carrying C. burnetii.

Line 200: cite the original reference after “China” (Yu et al., 2016, http://dx.doi.org/10.1016/j.actatropica.2016.02.027) as the sequence KT362172 is not described as “E. ruminantium-like” in Genbank. Also change the following sentence to “Further studies are needed to investigate the presence in bovine blood samples of the Ehrlichia sp. detected in the present study, and to assess its significance for animal health.” Yu et al found that their sequence was only 87.8% similar to E. ruminantium, which is not sufficiently close to identify either their sequence or the sequence obtained in the present study as E. ruminantium, or even (in this reviewer’s opinion), E. ruminantium-like. See also general comments on this point above.

Line 202: change to “Three species of Rickettsia were confirmed by phylogenetic analysis based on the Rickettsia 16S rRNA, gltA and ompA genes in this study…”

Line 210: delete “for the first time” – it contradicts the statement in the previous sentence.

Lines 237-251: this paragraph could be shortened – see general comments on co-infections.

Lines 238-239: change to “these microorganisms originated from ticks or from the host blood since…”

Lines 257-260: As the authors did not receive permission from the owners to collect blood samples from the cattle (which might anyway have required ethical approval from a Turkish institute or government body), it would be better to say “Although we were aware of the importance of this, it was outside the scope of the present study.”

Line 275: change to “Seventy apparently healthy cattle…”

Line 276: change to "severe tick infestations (1-45 ticks per animal) were detected...". 

Lines 309 and 310: this is not the first mention of B. bovis and T. orientalis

Author Response

Dear Editor and reviewers,

We are very grateful for your consideration of our manuscript. We have amended the corresponding content, which were highlighted in red in the revised manuscript, according to the comments from the reviewers. The detailed corrections in the manuscript and the response to the reviewers’ comments are shown in the file “Response to Reviewers”.

Reviewer 2

The authors have made a good effort to address most of the points raised by the reviewers. However, in this reviewer’s opinion, several points still require improvement to strengthen and bring the manuscript up to publication standard. This reviewer apologises for not raising some of these points during the first review round.

Response: We are very grateful for your consideration of our manuscript. The comments you provided are valuable and shall improve the quality of our manuscript.

  1. The issue of detection rate/prevalence can be addressed most usefully by determining the minimum detection rate for each microorganism in each tick species. This overcomes the difficulty in determining detection rate or prevalence when using pooled tick samples. Once calculated, this information should be presented in a Table and throughout the Abstract and text to replace “infection rate” or "prevalence". Calculating the minimum detection rate for a microorganism/tick species combination is done as follows, using the example of the combination ofTheileria annulata and Hyalomma excavatum. A total of 203  excavatum ticks were collected (line 68); from Table 2, the number of T. annulata-positive H. excavatum pools is 11. This means that a minimum of one tick in each of the 11 pools was positive for T. annulata (regardless of the number of ticks in the pool, and there is no way of knowing whether more than one tick was positive in any single pool). The minimum detection rate for T. annulata in H. excavatum is therefore [11 x 100] ÷ 203 = 5.4%. 

Response: We thank the reviewer’s suggestions. We have removed the old Table 1 and replaced it with a new table containing the information of minimum detection rate for each microorganism.

  1. In this reviewer’s opinion, the new Table 1 does not serve any useful purpose, as Table 2 has also been revised to present the results of microorganism detection by province. Although another reviewer asked for confidence limits and odds ratios to be included, the purpose of that information is not clear to this reviewer and it does not add anything to the interpretation of the data (the study is not about the effectiveness of the detection methods used, nor a comparison between different provinces or tick species; it is a descriptive study).

Response: Based on the reviewer's comments, we decided to delete Table 1 as the reviewer suggested.

  1. The term “co-infection” should only be applied to samples containing DNA from a single tick. If there is more than one tick in a pool, there is no way of knowing which detected microorganisms were in each of those ticks, and whether any tick was carrying more than one microorganism. Therefore, and bearing in mind the points raised above about minimum detection rate, this reviewer suggest that the new Table 1 is discarded, and Table 2 is amended to contain columns labelled “Province”; “Tick Species (Positive pools/Total pools)”; “Microorganism Detected (No. of pools)”; “Minimum Detection Rate”. The first row would then read (from left to right) “Gaziantep”;Hyalomma excavatum (8/X)”; “Theileria annulata (4)”; “5.4%” (where X is the total number of  excavatum pools from Gaziantep). The second row would contain the equivalent data for Babesia bovis detected in pools of H. excavatum from Gaziantep, and so on. If any pool containing only one tick was positive for more than one microorganism, this can be reported in the text as a co-infection, and discussed as such.

Response: We have removed column of co-infection and added the total pool nos. of each tick species. Likewise, we have calculated the minimum detection rate in the new Table 1.

  1. In Figure 1, the number of turanicus ticks collected in Sanhurfa is given as three, but in Table 2 there are four positive R. turanicus pools (each presumably containing at least one tick!), including one pool positive for both Ehrlichia sp. and T. annulata. This should be checked and corrected (either there should be more ticks or fewer pools).

Response: We appreciate the reviewer for pointing out this mistake. We have corrected it in Table 1.

  1. This reviewer questions whether or not one tick per cow can be considered as a "severe" infestation (authors' response to reviewer #3).

Response: We fully agree with reviewer. Indeed, one tick per cow cannot be considered as a "severe" infestation. We have removed "severe". Line: 269.

  1. Abstract, text and Tables 2 and 3 – ALL species names should be in full the first time they are mentioned in all of these locations.

Response: We have corrected it.

  1. This reviewer strongly recommends that the authors screen theEhrlichia-positive DNA samples with an Ehrlichia-specific 16S rRNA PCR and sequence at least one PCR product from each tick species. In Genbank there are 26 times more 16S rRNA sequences than pCS20 sequences deposited for Ehrlichia , so this gene should provide more information about the species present in the Turkish ticks. Although it is unlikely that E. ruminantium is present in Turkey, it is important from an animal health perspective to exclude it and, if possible, identify what Ehrlichia species is/are present and if they pose a risk to livestock.

Response: We fully agree with reviewer. We tried to answer and solve all the question or concern from reviewers. However, we currently do not have an Ehrlichia-specific 16S rRNA primer set. Due to the limited time for replying to the comments of the reviewers, we could not finish the required experiments within 7 days. Rest assured that we will continue the experiments to obtain the 16S rRNA sequence of the Ehrlichia-positive samples. In the meantime, we included it in the recommendations for future studies. Lines: 199-201.

  1. Lines 228-230: theEhrlichia  is very unlikely to be an endosymbiont. As far as this reviewer knows, there are no reports of Ehrlichia spp. behaving as symbionts, and they are not normally transovarially transmitted (except possibly in the case of the one-host tick R. microplus, see DOI 10.1186/s13071-016-1651-x). Ticks acquire Ehrlichia by feeding on an Ehrlichia-infected host.

Response: We appreciate your feedback. Following the advice of the reviewer, we have excluded Ehrlichia sp. in the discussion section about endosymbionts and just focused on Coxiella instead.

  1. Tables S1 and S2 – ALL species names should be in full, so in Table S1Hyalomma excavatumHyalomma anatolicum and Rhipicephalus bursa should all be in full, and in Table S2 Theileria ovisTheileria annulataBabesia bovis etc should all be in full. In Table S2, there should be an indication of which primer is the forward and which is the reverse for each pair, either in front of each primer, or by stating in the Table title that the forward primer is listed first in each case. Table S2 title should be change to “Primer sets used…”

Response: As per the reviewer’s suggestion, we have corrected it.

Specific points:

  1. Lines 23-24: change to “Diseases caused by tick-transmitted pathogens including bacteria, viruses and protozoa are of veterinary and medical importance especially in…”. Diseases do not adversely affect medicine, they adversely affect the humans and animals suffering from disease.Rickettsia are alphaproteobacteria belonging in the domain “Bacteria”; there is no need to list them separately from bacteria.

Response: We thank the reviewer for the suggestion. We have corrected it. Line: 23-24.

  1. Lines 33-34: change prevalences to minimum detection rates as explained in general comments below, and using the figures calculated in the specific comment on lines 87-90 below.

Response: We have modified it. Lines: 32-34.

  1. Line 36: change to “All sequences obtained from…”

Response: We have corrected it.

  1. Line 39: change to “and expand the knowledge”

Response: We have corrected it.

  1. Lines 52-53: change to “(43 from the family Ixodidae and 8 from the family Argasidae).”

Response: We have corrected it.

  1. Line 54: delete “(Boophilus)” – otherwise your statement excludes ticks such as bursaR. turanicus and R. sanguineus.

Response: We agree with reviewer and we deleted it.

  1. Line 75: change to “Figure 1. Map of sample collection sites in Turkey showing numbers of cattle examined (no. infested/total) and numbers of ticks of each species collected.

Response: We have changed as reviewer’s suggestion.

  1. Lines 87-90: change to: “The most frequently-detected pathogens in this study (Table 2) were bovis (minimum detection rate 7.9%, 22 positive pools), followed by Ehrlichia sp. (minimum detection rate 7.2%,20 positive pools), T. annulata (minimum detection rate 5.8%, 16 positive pools), Coxiella sp. (minimum detection rate 3.3%, 9 positive pools), A. marginale and Rickettsia spp. (minimum detection rate 2.5%, 7 positive pools each) and B. occultans (minimum detection rate 0.7%, 2 positive pools).

Response: We appreciate the reviewer’s suggestion and we have corrected it. Lines: 82-87.

  1. Lines 103-105: which tick species was the sequenced sample obtained from? This should be stated. Also change to “TheCoxiella-like sequence (OL413002)…” – there is no evidence from your study, or from the original article (doi:10.1093/gbe/evv108) that this Coxiella is a pathogen, i.e. a disease-causing agent.

Response: We have modified our description in Lines: 96-100.

  1. Line 150: change to “Overall, seven tick-associated microorganisms, including…” – it is not known whether or not theCoxiella  is transmitted by ticks to hosts during feeding.

Response: We have changed as reviewer’s suggestion. Line: 147.

  1. Line 165: change to “was highly similar to”

Response: We have corrected it.

  1. Lines 181-188: insert a statement saying that more work is needed to determine whether or not any of the tick species sampled in the present study were carrying burnetii.

Response: We have included the requested statement. Lines: 186-187.

  1. Line 200: cite the original reference after “China” (Yu et al., 2016, http://dx.doi.org/10.1016/j.actatropica.2016.02.027) as the sequence KT362172 is not described as “ ruminantium-like” in Genbank. Also change the following sentence to “Further studies are needed to investigate the presence in bovine blood samples of theEhrlichia sp. detected in the present study, and to assess its significance for animal health.” Yu et al found that their sequence was only 87.8% similar to E. ruminantium, which is not sufficiently close to identify either their sequence or the sequence obtained in the present study as E. ruminantium, or even (in this reviewer’s opinion), E. ruminantium-like. See also general comments on this point above.

Response: We have cited the reference. In addition, we have incorporated the statement that the reviewer provided in Lines 199-201.

  1. Line 202: change to “Three species ofRickettsia were confirmed by phylogenetic analysis based on the Rickettsia 16S rRNA, gltA and ompA genes in this study…”

Response: We have changed it.

  1. Line 210: delete “for the first time” – it contradicts the statement in the previous sentence.

Response: We have deleted it.

  1. Lines 237-251: this paragraph could be shortened – see general comments on co-infections.

Response: We have omitted irrelevant sentences in the paragraph to shorten it.

  1. Lines 238-239: change to “these microorganisms originated from ticks or from the host blood since…”

Response: We have corrected it.

  1. Lines 257-260: As the authors did not receive permission from the owners to collect blood samples from the cattle (which might anyway have required ethical approval from a Turkish institute or government body), it would be better to say “Although we were aware of the importance of this, it was outside the scope of the present study.”

Response: We appreciate the reviewer’s suggestion and we have changed it. Line: 252-253.

  1. Line 275: change to “Seventy apparently healthy cattle…”

Response: We have corrected it. Line: 268.

  1. Line 276: change to "severe tick infestations (1-45 ticks per animal) were detected...". 

Response: We have changed as the reviewer suggested.

  1. Lines 309 and 310: this is not the first mention of bovis and T. orientalis

Response: We have corrected it.

Reviewer 3 Report

Good job guys. I think you solved most of the problems I rose. 
I give you this feedback to you

Line 48: add production after milk.
line 83: sorry to be a pain but gene markers have to go to the M&M (as in the previous version) or in the introduction but definetly not in the result paragraph (you decide where to put it better). 
Line 103: you first say that tou have 10% sample positive for Coxiella sp. but now you mentione Coxiella-like. This doesn't make sense, so try to reword it. The suffix "-like" would imply that doesn't belong to Coxiell genus but is similar, contradicting what you said before.

Author Response

Dear Editor and reviewers,

We are very grateful for your consideration of our manuscript. We have amended the corresponding content, which were highlighted in red in the revised manuscript, according to the comments from the reviewers. The detailed corrections in the manuscript and the response to the reviewers’ comments are shown in the file “Response to Reviewers”.

Reviewer 3

  1. Good job guys. I think you solved most of the problems I rose. 
    I give you this feedback to you

Response: We are very grateful for your comments and suggestions which improved our manuscript quality.

  1. Line 48: add production after milk.

Response: We have added it.

  1. line 83: sorry to be a pain but gene markers have to go to the M&M (as in the previous version) or in the introduction but definetly not in the result paragraph (you decide where to put it better). 

Response: We thank the reviewer for the suggestion. We have deleted it because we have similar description in M&M. Lines: 298-305.

  1. Line 103: you first say that tou have 10% sample positive for Coxiella sp. but now you mentione Coxiella-like. This doesn't make sense, so try to reword it. The suffix "-like" would imply that doesn't belong to Coxiell genus but is similar, contradicting what you said before.

Response: We thank the reviewer for point it out. We have changed it to Coxiella sp. throughout the manuscript.